# An Effective Approach to Improve the Thermal Conductivity, Strength, and Stress Relaxation of Carbon Nanotubes/Epoxy Composites Based on Vitrimer Chemistry

**DOI:** 10.3390/ijms23168833

**Published:** 2022-08-09

**Authors:** Yang Feng, Zhuguang Nie, Panhong Deng, Liping Luo, Xingman Hu, Jie Su, Haiming Li, Xiaodong Fan, Shuhua Qi

**Affiliations:** Shannxi Key Laboratory of Macromolecular Science and Technology, School of Chemistry and Chemical Engineering, Northwestern Polytechnical University, Xi’an 710071, China

**Keywords:** dopamine, stress relaxation, vitrimer, shape memory polymer

## Abstract

An effective method was developed to improve the interfacial interaction between Mutiwalled carbon nanotubes (MWCNTs) and epoxy matrix. The performance of thermal conductivity and strength of the epoxy vitrimer were enhanced by polydopamine (PDA) coating. Polydopamine is a commonly used photothermal agent, which of course, was effective in modifying MWCNTs used in photoresponsive epoxy resin. The surface temperature of the epoxy composite with 3% MWCNTs@PDA fillers added increased from room temperature to 215 °C in 48 s. The metal–catechol coordination interactions formed between the catechol groups of PDA and Zn^2+^ accelerated the stress relaxation of epoxy vitrimer. Moreover, the shape memory, repairing, and recycling of epoxy vitrimer were investigated. Therefore, dopamine coating is a multifunctional approach to enhance the performance of epoxy vitrimer.

## 1. Introduction

Epoxy resin is a kind of crosslinking network thermoset polymer that has been widely used with excellent mechanical performance, processability, solvent resistance, and so on. However, it is not that possible to repair or reprocess epoxy materials once their permanent crosslinked structure is formed, which limits their application and adds a burden to the environment [1]. A new class of crosslinked polymeric material, known as vitrimers, has emerged with much attention in the area of materials science, with the combined advantages of both thermosets and thermoplastics [2,3]. Vitrimers are crosslinked polymers with dynamically exchangeable covalent bonds that can rearrange thermally while maintaining the integrity of the crosslinked network [4]. Thus, they can be processed repeatedly as thermoplastics at a temperature above T_v_ (the topology-freezing transition temperature) [5]. Much effort has been devoted to preparing vitrimer-based composites since then [6]. Qiao et al. prepared a liquid crystalline vitrimer containing oligoanilines, and it responded to six different stimuli and performed six functions [7]. Graphene, with the structure of a single layer of carbon atoms, is often used as a nanoscale heat source to increase the temperature of nanocomposites because of its highly-efficient photothermal transformation performance [8]. The carbon nanotube (CNTs)–vitrimer composite presented by Yang et al. was used to realize epoxy welding by light in seconds or minutes [9].

Despite this, the dispersion of the nanomaterials in the polymer matrix is very poor due to the high van der Waals forces and large aspect ratios, such as CNTs and graphene. This is the main problem for the reinforcement of mechanical properties in polymer composites [10,11]. Much effort has been made to enhance the dispersion of MWCNTs in matrices and the solubility of MWCNTs in common solvents. The relevant methods can be divided into two categories, namely covalent functionalization [12,13] and non-covalent functionalization [14,15]. Among the various modification methods, polydopamine (PDA) modification has been demonstrated as an effective approach to solving the problem of interfacial interactions between the nanomaterials and polymer matrix [16,17].

The polymerization process of dopamine is facile and simple. Nevertheless, the mechanism behind the formation of polydopamine is unclear and complex, which is caused by the generation of intermediates during the polymerization and reaction process [18]. Polydopamine can be used in many areas due to its properties in optics, electricity, magnetics, and biocompatibility. The functional groups incorporated into the polydopamine chemical structure, such as catechol, amine, and imine, have endowed it with many valuable features [19]. These functional groups can be used as active sites for covalent modification with molecules and anchors for the loading of transition metal ions. Additionally, it is the self-polymerization of dopamine that is regarded as the cause of pronounced absorption extending from visible to NIR wavelengths [18].

With efficient photothermal properties, PDA modification is the perfect method to solve the problem of interfacial interaction between nanomaterials and polymer matrix while maintaining high photothermal efficiency simultaneously. Herein, we provide a promising methodology to fabricate carbon nanotube/epoxy composites based on vitrimer chemistry with superior thermal conductivity, strength, and stress relaxation. The MWCNTs@PDA hybrids were first synthesized by a rational surface coating method, and the carbon nanotube/epoxy composite was prepared by compression molding. Figure 1 present the chemical structures of MWCNTs@PDA and epoxy composites crosslinked by dynamic transesterification and catechol–metal coordination. Furthermore, the influence of MWCNTs@PDA on the morphology, thermal conductivity, stress relaxation, and strength of the epoxy composites was investigated. The combined shape memory and dynamic covalent network mechanisms have also been comprehensively explained.

## 2. Results and Discussion

### 2.1. Characterizations of MWCNTs@PDA Hybrids

The polymerization of dopamine monomers immediately occurred under alkaline conditions in the buffer solution, coupled with a color change from colorless to brown and then black. Both covalent linkage and π–π stacking interactions are included in the connection between polydopamine and carboxylic MWCNTs, while only π–π interactions are present between the MWCNTs and PDA layer for pure MWCNTs [20]. Therefore, carboxylic MWCNTs were chosen as raw materials to prepare MWCNTs@PDA. Appendix A show the TEM images of MWCNTs@PDA with different thicknesses of the PDA layer. PDA layers were coated on the surface of MWCNTs by the polymerization of dopamine and a uniformly formed core–shell structure. The thickness of the PDA layers increased from 5 nm to 21 nm, with the mass ratio of dopamine monomer and MWCNTs increasing from 0.5:1 to 2:1. The thickness of the PDA coating can be controlled by tuning the mass ratio of dopamine and MWCNTs. The element mapping of MWCNTs and MWCNTs@PDA is displayed in Appendix A. It can be seen that C, N, and O elements were uniformly distributed on the surfaces of MWCNTs after PDA coating. Thus, the MWCNTs@PDA hybrids were prepared successfully. After PDA modification, the amide-rich surface of MWCNTs had the potential to form aggregates via hydrogen bonding during the curing reaction, which would restrict their dispersion [21]. In this study, the thicknesses of PDA coated on the surface of MWCNTs were controlled by the concentration of dopamine.

The dispersion of MWCNTs and MWCNTs@PDA in water solutions is shown in Appendix A. MWCNTs and MWCNTs@PDA were first homogeneously dispersed in water. After stewing for 30 days, the MWCNTs@PDA solution was still uniformly dispersed without any sedimentation. The hydroxyl groups and amine groups of PDA made MWCNTs@PDA exhibit better dispersibility than MWCNTs in water. 

More evidence of the presence of PDA structure can be proved by FT-IR analysis (Figure 2E). MWCNTs@PDA showed the characteristic peaks at 3453, 1643, and 1547 cm^−1^, which correspond to the OH, indole, and indoline groups of PDA. This further indicated that PDAs have successfully coated the surface of MWCNTs [22]. 

The XRD patterns of MWCNTs and MWCNTs@PDA are shown in Figure 2F. The broad diffraction peak at 26.3° corresponds to the (002) plane of MWCNTs, echoed by the rod−shaped graphitized MWCNT crystals. It can be observed that no crystalline peaks were generated with the introduction of PDA in MWCNTs, which confirms that no PDA existed as a single phase in MWCNTs@PDA. Figure 2A–D show the high−resolution C 1s XPS spectra of MWCNTs and MWCNTs@PDA. The C 1s spectra of MWCNTs can be deconvoluted into four peaks in Figure 2D. The peaks centered at 284.49, 284.95, 286.11, and 288.83 eV can be attributed to the C=C/C-C, C-O, C=O, and O-C=O groups, respectively. For MWCNTs@PDA hybrids in Figure 2B, compared with MWCNTs, a new peak centered at 285.54 eV can be assigned to the C-N group, which is due to the presence of PDA coatings [23]. In Figure 2A, the wide scan XPS spectrum indicates the existence of C, N, and O elements in the composites.

Appendix A show the weight loss during the heating process of MWCNTs, PDA, and MWCNTs@PDA in nitrogen. At temperatures up to 800 °C, the weight loss of MWCNT, PDA, and MWCNTs@PDA is 11.15%, 15.88%, and 50.19%, respectively. By comparison with MWCNTs, the thermal stability of MWCNTs@PDA is reduced. On the basis of the weight loss, it is calculated that MWCNTs@PDA contains 12.12% PDA. 

### 2.2. Dispersion of MWCNTs@PDA Hybrids in the Epoxy Matrix

The thermal conductivity and mechanical properties of nanocomposites are closely associated with the dispersion of nanomaterials. The dispersibility of CNTs in the polymer matrix is very poor due to their high van der Waals forces and large aspect ratio. The high intrinsic thermal conductivity of CNTs can be significantly decreased by simple mixing into the matrix, which is mainly caused by defects and phonon scattering [24]. 

The dispersion states of MWCNTs and MWCNTs@PDA hybrids in the epoxy composites are shown in Figure 3. It was observed that the pristine MWCNTs dispersed poorly in the epoxy matrix. Nevertheless, the PDA-modified MWCNTs were homogeneously dispersed in the matrix, which is caused by the incorporation of the functional groups such as catechol, amine, and imine. As described in Figure 1, in the process of PDA coating, the amine groups of PDA could be covalently connected with the carboxyl groups of MWCNTs. Furthermore, the interfacial interactions between MWCNTs and epoxy matrix were enhanced by the combined mechanisms of the hydrogen bonding between the amino group and hydroxyl group existing in PDA, the abundant hydroxyl groups of epoxy resin, and polymer entanglement between epoxy and PDA [21]. 

### 2.3. Thermal Conductivity and Mechanical Properties of MWCNTs @PDA/Vitrimer Composites

Figure 4a show the thermal conductivity of MWCNTs@PDA epoxy composites at room temperature. For all samples, the thermal conductivity of MWCNTs@PDA epoxy composites increased with the addition of MWCNTs. The thermal conductivity of epoxy without any filler was 0.16 W/mK, and it increased to 0.28 W/mK and 0.34 W/mK with 3.0 wt% MWCNTs and 3.0 wt% MWCNTs@PDA(0.5:1) added, respectively. All the mass fractions of fillers and the thermal conductivity of epoxy composites with MWCNTs@PDA(0.5:1) added were higher than the epoxy composites with MWCNTs@PDA(1:1), and MWCNTs@PDA(2:1) added. It could be explained that the thickness of MWCNTs@PDA(0.5:1) was thinner than others, which could decrease phonon scattering during the heat transfer process. Meanwhile, the thermal conductivity of epoxy nanocomposites was also strongly related to the dispersion state of the fillers in the matrix. Compared with the other two samples, fewer amino groups existed at the surfaces of MWCNTs@PDA(0.5:1), which could avoid the aggregation of MWCNTs. Thermal triggering is one of the most commonly used methods to initiate self-healing behaviors. Even for the NIR triggering method, it was essentially a process of photothermal transfer.

The typical stress–strain curves at room temperature are shown in Figure 4b, and the data of tensile strength and tensile modulus are summarized in Table 1. The tensile strength and tensile modulus were enhanced when the mass fraction of MWCNTs increased from 0 % to 1.0 wt%. The reinforcement effect is a complex issue involving load transfer, stress concentration, and defect distribution [8]. Further increase of the MWCNTs content could have an adverse effect on tensile strength. It could be explained that MWCNTs could increase the number of microcracks, which can limit the propagation of the cracks. Furthermore, the well-dispersed MWCNTs@PDA in the epoxy matrix and strong interfacial bonding could enhance the tensile strength [25]. The polar groups such as catechol and imine groups on the surfaces of MWCNTs@PDA could strongly enhance the interfacial interactions, thus bringing more effective stress transfer from high modulus MWCNTs to the epoxy matrix [26]. The tensile strength of EP m and EP-M@PDA decreased with the mass fraction of fillers increasing from 1.0 to 3.0 wt%, which could be attributed to the introduced defect and poor dispersion of fillers. Combined with the SEM images of fracture surfaces in Figure 5, it can be found that the pure epoxy displays a smooth and flat fracture surface, exhibiting a typical brittle fracture feature. No events can be observed during the crack propagation due to the lack of energy-absorbing events. Compared with pure epoxy, the introduction of the nanofillers produced a slightly disordered and rough surface, which was caused by the presence of MWCNTs. The MWCNTs could interrupt the cracking direction and restrain the propagation of the cracks, making the cracks sequentially distributed in a disorderly way [26]. The existence of MWCNTs is beneficial for stress transfer from epoxy to high modulus MWCNTs. However, the agglomeration of MWCNTs in epoxy composites during the curing process caused heterogeneous dispersion, resulting in stress concentration. The fracture surface of MWCNTs@PDA epoxy composites became rougher, and the cracks became much more disorderly. This may be attributed to the anchored catechol on the surfaces of MWCNTs, which could increase the interfacial interaction between MWCNTs and the epoxy matrix. Meanwhile, the amine groups on the surfaces of MWCNTs can be connected with the epoxide group in the polymer chain via covalent bonding. The method of PDA coating improved the interfacial interaction between MWCNTs and epoxy, induced more effective stress transfer from the polymer matrix to MWCNTs, and made the fracture surface rough. 

### 2.4. Thermal Stability and Dynamic Mechanical Properties of MWCNTs @PDA/Vitrimer Composites

The thermal stability of MWCNTs@PDA epoxy composites was evaluated by TGA analysis under a nitrogen atmosphere in Figure 6. It can be observed that there is almost no distinction between the TGA curves of epoxy and epoxy composites. The initial decomposition temperatures (~5 wt% weight loss) of all samples are summarized in Table 1. The initial decomposition temperature of epoxy composites increased with the addition of MWCNTs@PDA fillers (from 308.5 °C for pure epoxy to 320.4 °C for 3 wt% loading of MWCNTs@PDA epoxy composites). The fastest weight loss occurs at almost the same temperature (~400 °C). Thus, the addition of MWCNTs@PDA has no impact on epoxy composites. 

The glass transition temperatures (T_g_) of the epoxy composites were characterized by DSC (Figure 7), and the corresponding data are shown in Table 1. With the addition of MWCNTs@PDA, T_g_ of the epoxy composites increased accordingly. The glass transition temperature refers to the temperature at which the network polymer chain begins to move. The movement of network segments can be influenced by chemical crosslinking, physical entanglement, and packing density of the segments. The MWCNTs@PDA occupied part of the free volume of the epoxy composites and hindered the movement of the polymer chain. Meanwhile, the chemical interaction between MWCNTs@PDA and the epoxy chain can restrict the movement of network segments [8,27].

The dynamic mechanical properties of MWCNTs@PDA/epoxy composites were studied by DMA in Figure 8a. The storage modulus of all samples decreased by three orders of magnitude between the glass state modulus and the rubber state modulus while heating through the glass transition temperature (T_g_), exhibiting similar temperature-dependent viscoelastic properties and indicating their potential shape memory behaviors. The storage modulus of neat epoxy at the glass state is about 1680 MPa. The presence of a chemically crosslinked polymer network by a rubber platform could reach 3.4 MPa at 90 °C. In addition to the differential scanning calorimetry method, dynamic mechanical analysis could also be used to obtain the glass transition temperature of polymers. Figure 8b show the relationship between the value of tanδ and temperature. T_g_ increased with the addition of MWCNTs@PDA. The T_g_ values of all samples are shown in Table 1. The T_g_ values determined by the DMA were higher than those determined by the DSC, which can be attributed to the differences in the measuring method.

### 2.5. Stress Relaxation Behavior of MWCNTs@PDA Epoxy Composites

Figure 9 show the evolution of the relaxation modulus at 180 °C for the epoxy composites synthesized with different content of MWCNTs and MWCNTs@PDA hybrids. Obviously, stress relaxation occurred at 180 °C for all the samples, indicating that the epoxy network can flow even though incorporated with high-loading fillers. It is attributed to the presence of the transesterification catalyst zinc acetylacetonate. Zn^2+^ acted as a catalyst by activating ester carbonyl, stabilizing alkoxide groups and moving them close to each other, thus catalyzing the exchange reactions in epoxy vitrimers [28]. 

However, epoxy composites exhibit much slower relaxation than the epoxy vitrimer matrix. MWCNTs are one-dimensional nanofillers with a large length–diameter ratio. The coiling of MWCNTs retarded the interfacial transesterification reaction in the epoxy network, which impaired the rate of relaxation [29]. The catechol groups of PDA anchored partial Zn^2+^ of the catalyst zinc acetylacetonate, which retarded the exchange reactions. Notably, the stress relaxation of MWCNTs epoxy composites occurs more slowly than the MWCNTs@PDA epoxy composites. The relaxation times of EP, EP-M-0.5%, and EP-M@PDA-0.5% were 247.0 s, 294.5 s, and 447.6 s, respectively. For MWCNTs@PDA epoxy composites, the amino groups on the surface of MWCNTs can react with epoxide groups. Thus, MWCNTs can be connected to the epoxy matrix by covalent bonding. Meanwhile, large amounts of catechol existed on the surface of MWCNTs and could be linked with Zn^2+^ by metal–catechol crosslinking interactions. The dynamic reversible metal–catechol coordination bonds can also participate in the relaxation process of the whole material [30,31]. Therefore, the metal–catechol coordination interactions in the MWCNTs@PDA epoxy network can accelerate the stress relaxation of epoxy composites compared with the MWCNTs epoxy network. However, the acceleration of metal–catechol coordination was limited and only applied to epoxy vitrimer with metal catalysts such as zinc acetylacetonate and zinc acetate. 

### 2.6. Shape Memory and Repairing of MWCNTs@PDA Epoxy Composites

The shape memory, repair, and recycling of epoxy vitrimers may be caused by the stress relaxation induced by the dynamic transesterification exchange reaction. As Figure 10 show, in classical epoxy/carboxylic acid polymer networks, the abundance of both the free hydroxyl functional groups and carboxylic ester functional groups can be guaranteed by simply mixing the stoichiometric amounts of bi-and poly-functional monomers. The mechanism of transesterification belongs to the associative exchange mechanism, and the covalent adaptable networks guarantee the materials to relax stresses and flow. Among them, covalent bonds can only be broken when new ones are formed, making these networks permanent as well as dynamic [2]. The reversible reaction between the epoxy group and carboxylic acids results in repeating units, which can be controlled by heat. 

To investigate the shape memory properties of epoxy composites, EP-M@PDA-1.0% was tested to obtain the shape fixity ratio (R_f_) and shape recovery ratio (R_r_) with the strain above 22.5% and the recovery temperature at 70 °C. As shown in Figure 11, for the first shape memory cycle, the values of R_f_ and R_r_ were about 80.1% and 92.1%, while for the second shape memory cycle, the values of R_f_ and R_r_ were about 82.9% and 92.1%. 

Figure 12 show the shape memory behavior of EP-M@PDA-1.0% with a strip sample, a spiral sample, and a u-type sample. The originals were strip samples, and they were configured at 200 °C for 2 h under external force. The strip samples were deformed to a permanent spiral shape at 70 °C, which was above the glass transition temperature, and then cooled to room temperature. After heating at 70 °C, the shape of the permanent spiral shape recovered to a strip shape.

The spiral samples were deformed into permanent strip shapes at 70 °C, which was above the glass transition temperature, and then cooled to room temperature. After heating at 70 °C, the shape of the permanent strip recovered to a spiral shape. The recovery process of the spiral sample could occur in less than one minute. The u-type sample deformed to a permanent strip shape at 70 °C, which was above the glass transition temperature, and then cooled to room temperature. After heating at 70 °C, the shape of the permanent spiral recovered to a u-type.

With the excellent properties of photothermal conversion, the shape recovery could be induced by light, such as near-infrared light (NIR). The shape recovery process of EP-M@PDA-1.0% induced by the NIR laser is shown in Figure 13. Both the deformation process and the recovery process were triggered by NIR. Partial areas of four flat films of EP-M@PDA-1.0% were irradiated by NIR (500 mW/cm^2^) and deformed to a temporary shape “N-W-P-U” after this treatment. The temporary shape can also recover to its original shape using NIR irradiation, and the recovery process could be completed in a few seconds. The shape recovery triggered by light is an accurate and efficient way to realize the process of shape memory.

To study the repairing properties of epoxy vitrimers, neat epoxy and EP-M@PDA-1.0% were scratched to have a crack on the surface. The repair process was also triggered by NIR (1.5 W/cm^2^). The surface temperature changes of MWCNTs@PDA epoxy composites under NIR irradiation (1.5 W/cm^2^) were recorded by the infrared camera (Figure 14A). The temperature of MWCNTs@PDA epoxy composites increased much faster than that of the neat epoxy. No NIR photothermal conversion properties were observed for neat epoxy, and the surface temperature of neat epoxy was stable at near 23 °C even after the 40 s irradiation. With the addition of MWCNTs@PDA to the epoxy, the surface temperature of the composite increased much faster and higher than neat epoxy. The temperature of the epoxy composite with 3% MWCNTs@PDA fillers added increased to 215 °C in 48 s, which exceeded the initial temperature of the transesterification exchange reaction. Therefore, the crack of neat epoxy still existed on the surface, while the crack of MWCNTs@PDA disappeared, as shown in Figure 14B. This can be attributed to the temperature of EP-M@PDA-1.0% reaching the initial temperature of transesterification exchange reaction and covalent reaction exchanged quickly with the promotion of catalyst zinc acetylacetonate.

### 2.7. Recycling of MWCNTs@PDA Epoxy Composites

In some cases, where materials are seriously damaged and out of use, recycling is necessary. It is almost impossible to recycle conventional thermosetting polymers since they form constant crosslinks. The incorporation of dynamic bonds into the networks provides a satisfying solution for polymer recycling. To demonstrate the reprocessing ability of the prepared epoxy composites, samples were cut into particles and reprocessed by hot-pressing under the pressure of 15 MPa at 180 °C for 2 h. Tensile strength testing was used to evaluate the effect of reprocessing on the mechanical properties of all samples. As shown in Figure 15, the tensile strength of the EP, EP-M-0.5%, and EP-M@PDA-0.5% after reprocessing could be maintained at 90.0%, 89.7%, and 89.3% compared to that of the samples before recycling, respectively. All samples show no difference in recycling efficiency. It can be seen that the samples can recover most of the mechanical properties after hot pressing, indicating the excellent recyclability of the covalently crosslinked epoxy composites.

## 3. Materials and Methods

### 3.1. Materials

Pristine multi-walled carbon nanotubes (MWCNTs), with a mean diameter of 8~15 nm and a length of 50 μm, sebacic acid, zinc acetylacetonate, were supplied by Macklin Biochemical Co., Ltd. (Shanghai, China). Tris(hydroxymethyl)aminomethane (Tris) and dopamine hydrochloride were purchased from Aladdin Biochemical Technology Co., Ltd. (Shanghai, China). Epoxy resin, diglycidyl ether of bisphenol A (WSR618), with an equivalent epoxy weight of 184–200 g(equiv)−1, was purchased from Nantong Xingchen Synthetic Material Co., Ltd. (Nantong, China). Other reagents were of analytical grade or better and used without further purification.

### 3.2. Preparation of MWCNTs@PDA

MWCNTs (200 mg) were dispersed in DI-H_2_O (20 mL) under ultrasonication for 30 min. Then, dopamine hydrochloride was added to the above solution at ratios of 2:1, 1:1, and 0.5:1 (dopamine hydrochloride: MWCNTs) under magnetic stirring, followed by 40 mL of tris-buffer solution (pH = 8.5). The whole mixture was stirred vigorously at room temperature for 12 h. Finally, the final product was filtered by a 0.2 μm membrane filter and washed with DI-H_2_O until the filtrate became colorless. Finally, MWCNTs@PDA hybrids were obtained by the freeze-drying method.

### 3.3. Preparation of MWCNTs @PDA/Vitrimer Composites

E51(5 g) was dissolved in 20 mL ethanol, and different amounts of as-prepared MWCNTs@PDA were dispersed in the above-mixed solution under ultrasonication for 10 min. The mixture was heated to 80 °C until the ethanol was totally evaporated. After that, the mixture and sebacic acid (SA, 2.58 g) were mixed at 140 °C under manual stirring. Then, zinc acetylacetonate (0.336 g, 5 mol% to the COOH) was introduced and stirred until homogeneously mixed. Finally, the mixture was poured into a stainless-steel mold and cured in 15 MPa at 140 °C for 1h and 180 °C for 4 h.

### 3.4. Characterization

The microstructure of the composites was characterized by field emission scanning electron microscopy (FESEM, FEI, Hillsboro, OR, USA, Verios G4, 10 kV, 0.1 nA). The crystal structure of all the samples was collected by X-ray diffraction (XRD, Shimadzu, Japan MAXima GLXRD-7000) at 40 kV and 100 mA of Cu Kα. The surface chemical compositions of MWCNTs@PDA were analyzed with an X-ray photoelectron spectrometer (PHI5000). The morphology of the synthesized samples and the dispersion qualities of the filler in epoxy composites were examined with transmission electron microscopy (TEM, FEI, Talos F200X TEM). The epoxy composites were mounted on the freezing microtome (UC7FC7, LEICA, Wetzlar, Germany) to obtain a 0.1 μm thick section. The section was transferred to a copper grid for close examination by TEM. The element mapping was obtained by EDS with transmission electron microscopy. Synchronous thermogravimetric (TG, München, Germany, STA449F3 Jupiter) was carried out with the temperature ranging from room temperature to 800 °C and a heating rate of 10 °C/min under air and nitrogen atmosphere. Differential scanning calorimetry (DSC) analysis was performed with a Mettler Toledo (New Port Richey, FL, USA) instrument under a nitrogen atmosphere with a heating rate of 10 °C/min. An infrared thermal imager (IRS, Suzhou China, IRS-S6, temperature range −20 °C–650 °C) was applied to characterize the photothermal process and to record the temperature change of the surface. Dynamic mechanical analysis (DMA) was performed on Netzsch DMA 242 E heating from −30 to 110 °C at a rate of 5 °C/min and a frequency of 1 Hz. Stress relaxation was performed on a Netzsch DMA 242 E. After thermal equilibrium at testing temperature, a strain of 5% was applied within the liner region of the materials, and the relaxation modulus was collected as a function of time. The shape memory behavior of samples was evaluated by DMA 242 E using a tensile method and at the controlled force mode. The thermal conductivity was determined by a hot disk instrument (AB Corporation, Västerås, Sweden).

## 4. Conclusions

In this study, thermal and NIR triggered shape memory composites were prepared by dynamic transesterification reaction. The approach of PDA coating significantly improved the mechanical properties and thermal conductivity of the composites, and the metal–catechol coordination interactions in the MWCNTs@PDA epoxy network were found to accelerate the stress relaxation of epoxy composites compared with the MWCNTs epoxy network. The relaxation time was 294.5 s (180 °C) for MWCNTs@PDA epoxy composites and 447.6 s (180 °C) for MWCNTs epoxy composites, respectively. Under NIR irradiation (1.5 W/cm^2^), the surface temperature of the epoxy composite with 3% MWCNTs@PDA added increased to 215 °C in 48 s, which initiated the transesterification exchange reaction and realized the healing of the crack rapidly. Overall, PDA coating may be a promising alternative to improve the interfacial interactions between the nanofiller and polymer matrix and may be a multifunctional approach to enhance the performance of epoxy vitrimers.

## Figures and Tables

**Figure 1 ijms-23-08833-f001:**
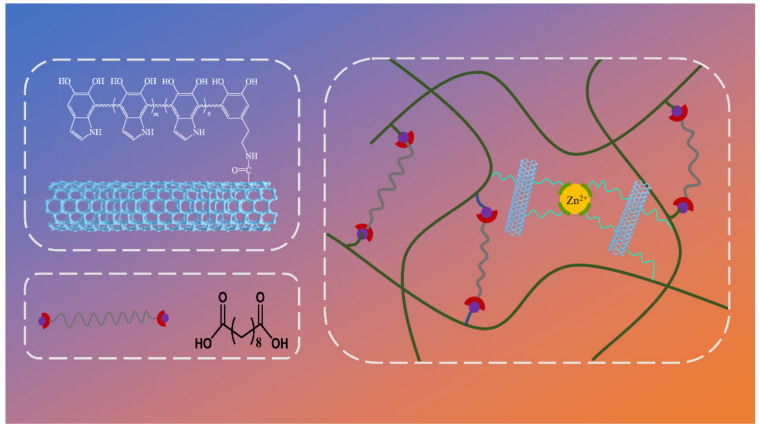
Chemical structures of MWCNTs@PDA and network of epoxy composites crosslinked by dynamic transesterification and metal–catechol coordination.

**Figure 2 ijms-23-08833-f002:**
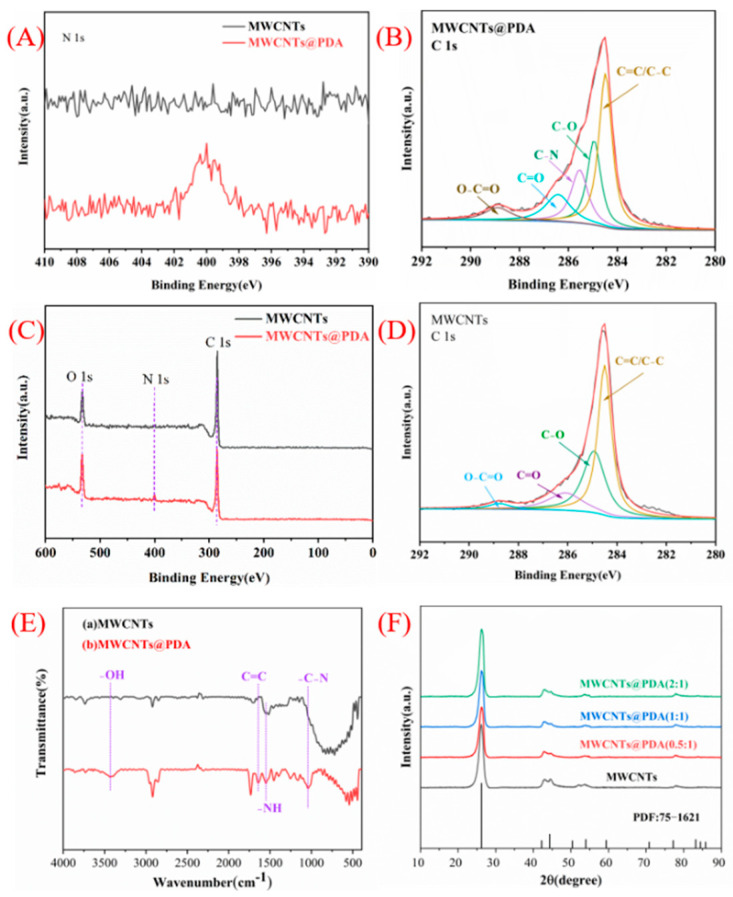
High−resolution XPS spectra of (**A**) N 1s peaks for MWCNTs and MWCNTs@PDA, (**B**) C 1s peaks for MWCNTs@PDA, (**C**) XPS spectra of MWCNTs and MWCNTs@PDA, (**D**) C 1s peaks for MWCNTs, (**E**) FT−IR spectra of (a) MWCNTs, (b) MWCNTs@PDA, (**F**) XRD patterns of MWCNTs and MWCNTs@PDA.

**Figure 3 ijms-23-08833-f003:**
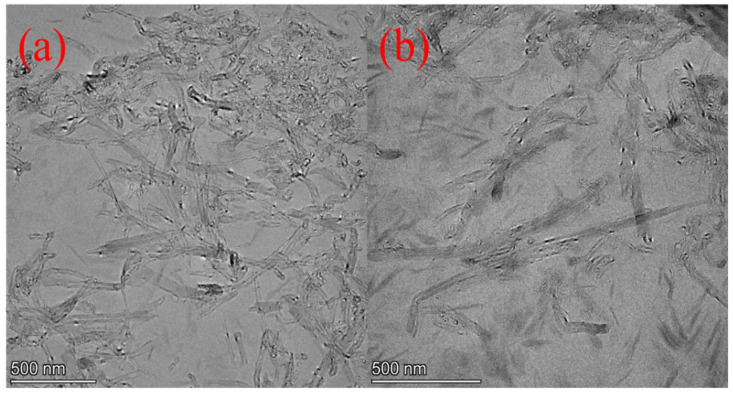
TEM images of (**a**) EP-M-1.0%. (**b**) EP-M@PDA-1.0%.

**Figure 4 ijms-23-08833-f004:**
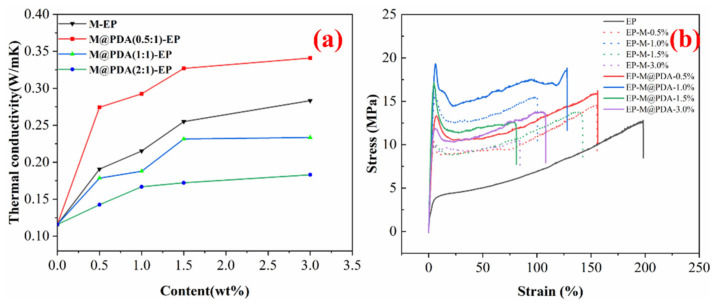
(**a**) Thermal conductivity of MWCNTs@PDA epoxy composites. (**b**) Typical stress–strain curves of MWCNTs epoxy composites and MWCNTs@PDA epoxy composites.

**Figure 5 ijms-23-08833-f005:**
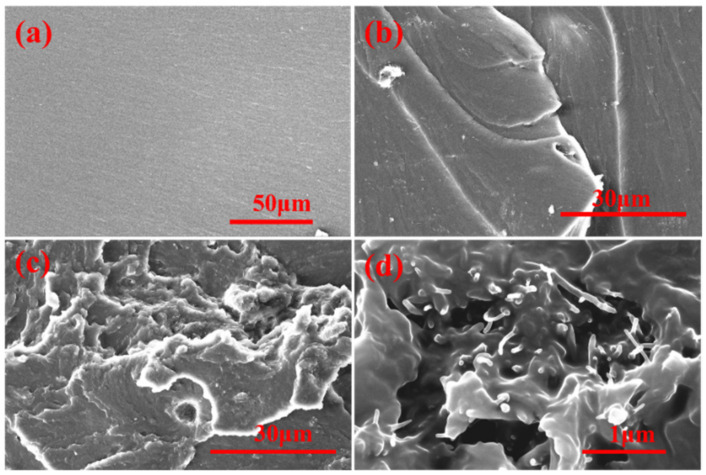
SEM images of fracture surfaces for (**a**) EP. (**b**) EP-M-3.0%. (**c**) and (**d**) EP-M@PDA-3%.

**Figure 6 ijms-23-08833-f006:**
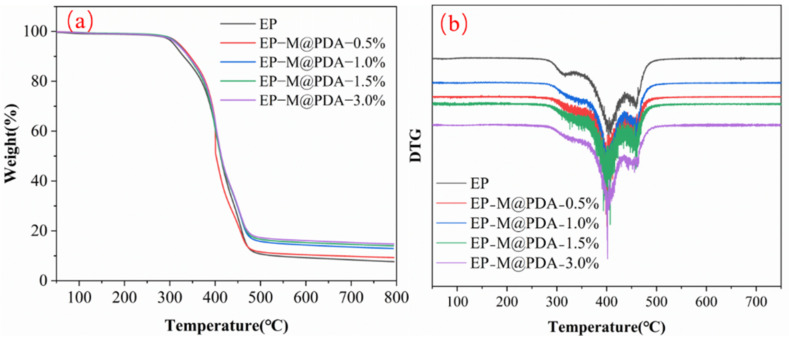
(**a**) Thermogravimetric analysis of the MWCNTs@PDA epoxy composites. (**b**) Derivative thermo-gravimetric (DTG) analysis of the MWCNTs@PDA epoxy composites.

**Figure 7 ijms-23-08833-f007:**
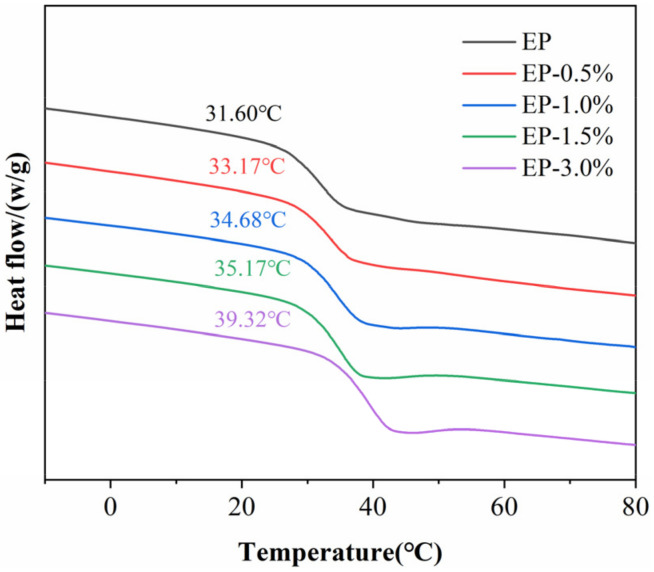
The differential scanning calorimeter (DSC) curves of MWCNTs@PDA epoxy composites (exo up).

**Figure 8 ijms-23-08833-f008:**
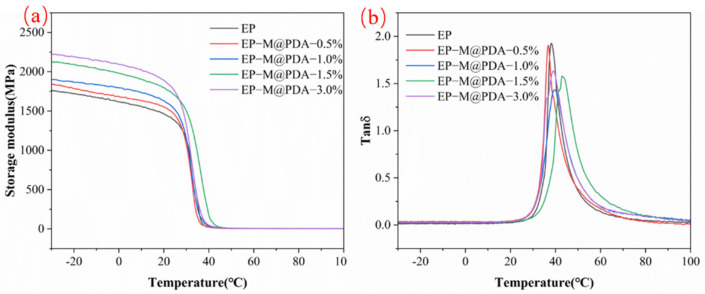
(**a**) Storage modulus vs. temperature curves. (**b**) Tan δ vs temperature curves.

**Figure 9 ijms-23-08833-f009:**
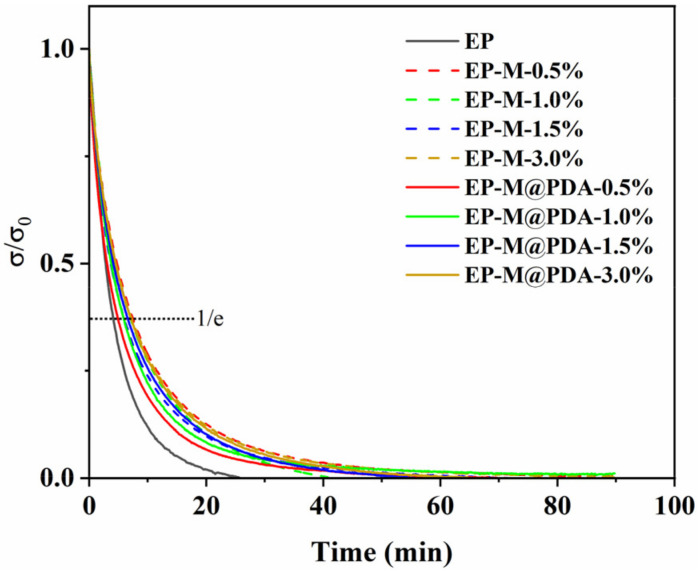
Stress relaxation curves of epoxy vitrimers at 180 °C.

**Figure 10 ijms-23-08833-f010:**
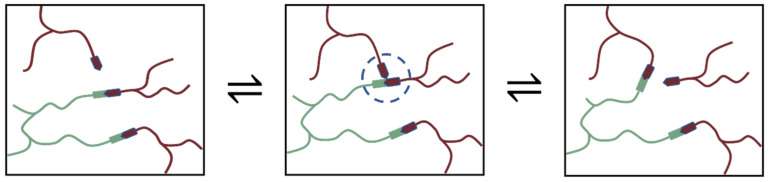
Epoxy vitrimers crosslinked by dynamic transesterification reaction.

**Figure 11 ijms-23-08833-f011:**
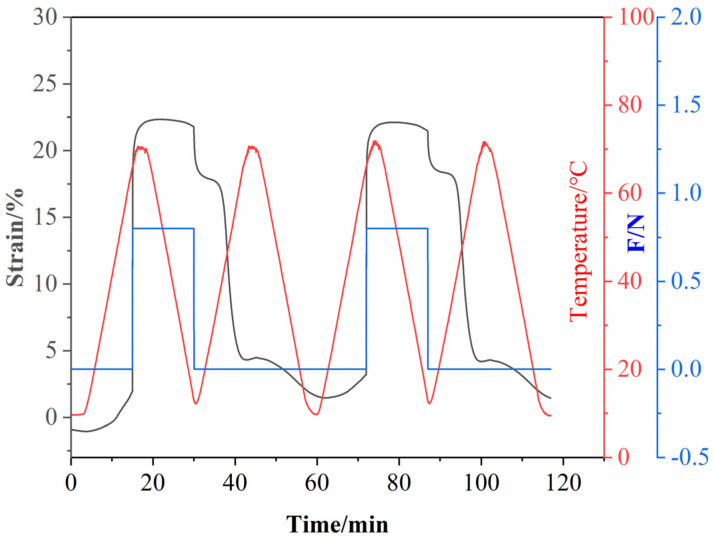
Two Shape memory cycles for EP-M@PDA-1.0%.

**Figure 12 ijms-23-08833-f012:**
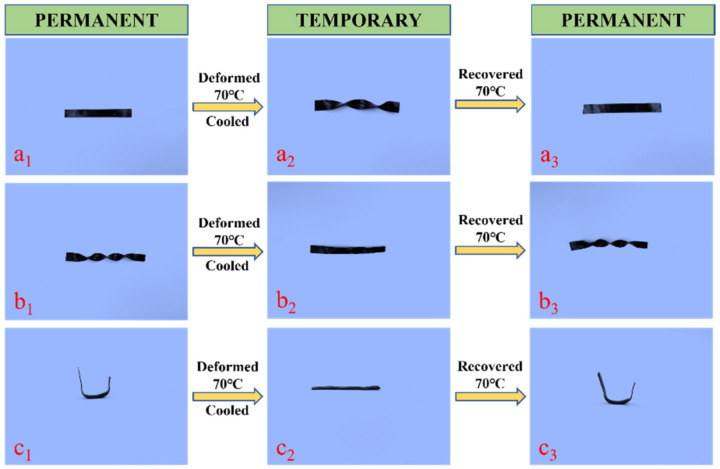
Shape memory behavior of (**a_1_**–**a_3_**) strip sample. (**b_1_**–**b_3_**) spiral sample (**c_1_**–**c_3_**) u-type sample.

**Figure 13 ijms-23-08833-f013:**
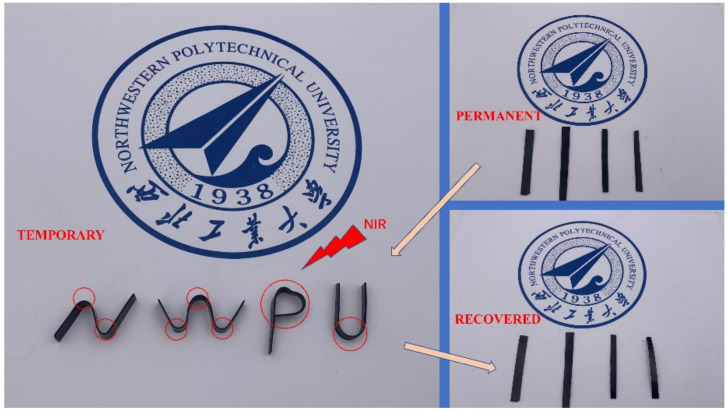
The shape memory process of EP-M@PDA-1.0% triggered by NIR.

**Figure 14 ijms-23-08833-f014:**
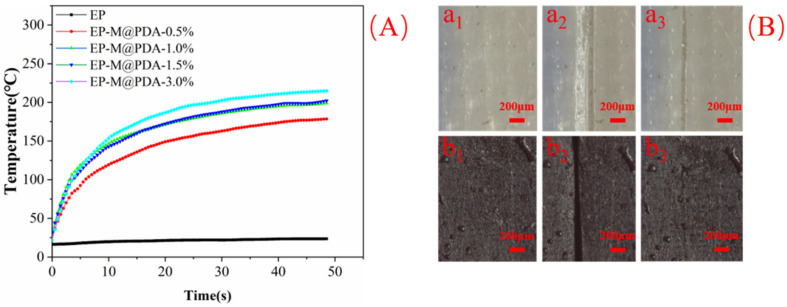
(**A**) Surface temperature change of epoxy composites with irradiation time. (**B**) Surface images of (**a_1_**,**b_1_**) images of original neat epoxy and EP-M@PDA-1.0% (**a_2_**,**b_2_**) images of neat epoxy and EP-M@PDA-1.0% scratched (**a_3_**,**b_3_**) images of neat epoxy and EP-M@PDA-1.0% after NIR irradiation for 2 min.

**Figure 15 ijms-23-08833-f015:**
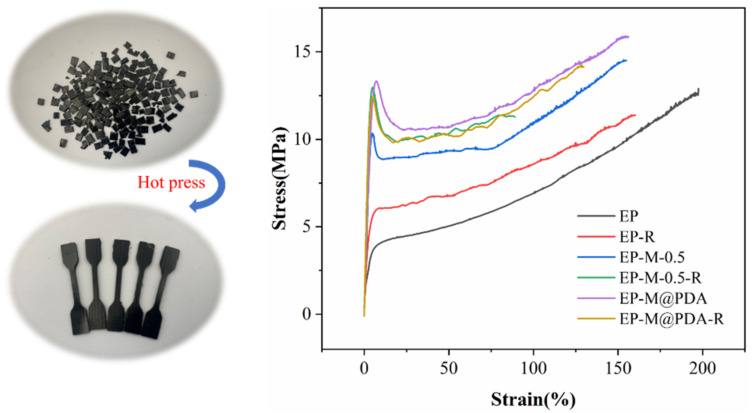
Typical stress–strain curves of virgin and reprocessed epoxy composites.

**Table 1 ijms-23-08833-t001:** Physical Properties of epoxy composites.

Sample	Tensile Modulus (MPa)	Tensile Strength (MPa)	Storage Modulus at Glass State (MPa)	Tg^DMA^ (°C)	T_5%_ (°C)
EP	100.65 ± 6.40	13.43 ± 0.56	1679.99	38.05	308.5
EP-M-0.5%	297.15 ± 6.05	13.33 ± 3.48	-		
EP-M-1.0%	384.45 ± 15.00	16.74 ± 1.18	-		
EP-M-1.5%	313.63 ± 7.59	15.08 ± 1.56	-		
EP-M-3.0%	349.11 ± 12.58	10.15 ± 0.72	-		
EP-M@PDA-0.5%	296.92 ± 3.41	16.33 ± 0.93	1737.49	36.72	315.3
EP-M@PDA-1.0%	472.07 ± 9.67	20.13 ± 1.64	1834.11	38.62	316.5
EP-M@PDA-1.5%	496.35 ± 15.03	18.43 ± 1.07	2041.72	43.13	317.6
EP-M@PDA-3.0%	412.53 ± 18.19	14.76 ± 2.07	2147.12	39.02	320.4

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
