# Peer review of "An Effective Approach to Improve the Thermal Conductivity, Strength, and Stress Relaxation of Carbon Nanotubes/Epoxy Composites Based on Vitrimer Chemistry"

_ijms, 2022, doi:10.3390/ijms23168833_

Round 1

Reviewer 1 Report

please, find attached file.

Reviewer 2 Report

In the reviewed work  authors focused on improving interactions between MWCNTs and epoxy matrix and on enhancing the thermal conductivity and strength of the epoxy vitrimer, i.e. a cross-linked polymer with dynamically exchangeable covalent bonds that can rearrange thermally while maintaining the integrity of the cross-linked network. As a photothermal agent polydopamine was used for modification of MWCNTs used in photo-responsive epoxy resin. Improved  mechanical properties and thermal conductivity of the composites, as well as enhanced stress relaxation behaviour have been observed in the obtained composite materials. The subject of this worki s interesting, however, there are a number of issues that need to be further addresed:

- the title contains the term „biomimetic approach” , however, it is neither explained nor evidenced in the course of experiments performed; simple use of  a biopolymer (polydopamine) is not enough.

- „Introduction” does not show recent important achievements in this field – please complete and expand.

-  Fig. 6 – what are the units for DTG?

- Fig. 7 – please indicate exo up or down?

- what is the thermal stability of MWCNT@PDA hybrids?

- How the (spiral) geometry of the samples influenced the shape memory behaviour?

- Abstract and Conclusions are rather poor – please re-write&expand.  

English needs correction by a native speaker, e.g. „Polydopamine is a commonly used photothermal agent, which of course, was an efficient method”, „The polymerization process of polydopamine” (dopamine?), „It is these functional groups that can be used…”, „Derivative thermos gravimetric…”, etc.

Surprisingly, an important referernce work  - https://doi.org/10.1016/j.pmatsci.2020.100710 -has not been cited.

Round 2

Reviewer 2 Report

Authors provided proper responses to reviewer's comments.  The revised manuscript can be accepted.